# Samarium-Mediated Asymmetric Synthesis

Suman Majee [1], Devalina Ray [1,2,*] and Bimal Krishna Banik [3,*]

1    Amity Institute of Click Chemistry Research and Studies, Amity University, Sector 125, Noida 201313, India
2    Amity Institute of Biotechnology, Amity University, Sector 125, Noida 201313, India
3    Department of Mathematics and Natural Sciences, College of Sciences and Human Studies,
     Prince Mohammad Bin Fahd University, Dhahran 34754, Saudi Arabia
*    Correspondence: dray@amity.edu (D.R.); bimalbanik10@gmail.com or bbanik@pmu.edu.sa (B.K.B.)

**Abstract:** Samarium is an efficient reducing agent, a radical generator in cyclization and a cascade addition reaction. Interestingly, samarium metal has crucial impact on numerous C-C and C-X (X = hetero atom) bond forming transformations. It has been established as an exceptional chemo-selective and stereoselective reagent. The reactivity of the samarium catalyst/reagent is remarkably enhanced in the presence of various additives, ligands and solvents through effective coordination and an increase in reduction potential. It has inherent character to act as electron donor for a wide range of transformations including the asymmetric version of various reactions. This review accentuates the developments in samarium-mediated/catalyzed asymmetric organic synthesis over the past 12 years, where the chirality has been induced from ligand, a nearby asymmetric center within the substrate or through coordination directed stereospecific reactions.

**Keywords:** samarium; asymmetric reactions; radical cyclization; cascade reactions





## 1. Introduction

Samarium, as a lanthanide metal, prefers to form low valent complexes and is most stable in the +3 oxidation state. It preferably acts as one electron mild reducing agent in its +2 oxidation state, which can be generated by reacting Sm(III) salts with metallic samarium [1–6]. Among all other salts of samarium, samarium(II) iodide ($SmI_2$), also known as Kagan's reagent, has exclusive properties to act as a powerful but selective reagent for a wide range of synthetic transformations following both the radical and ionic pathways [7–9]. The commercially available $SmI_2$ has been vastly utilized in organic reactions through the in situ generation of active organosamarium by one-electron or two-electron electron transfer pathways leading to the generation of radical or anion intermediates, mostly in coupling reactions [10,11]. Moreover, the role of the ligand, additive and solvents are very prominent; leading to notable enhancement in the reactivity of the samarium reagent [3,12–14]. The coordination sphere of samarium is crucial to estimate the reactivity of that catalyst. The oxyphilic nature drives it to bind with the oxygen containing solvents and additives, which imparts substantial impact on the coordination sphere of samarium [15–18]. Due to the high solubility of $SmI_2$ in tetrahydrofuran (THF), it is the commonly preferred solvent in this reaction. $SmI_2$ shows pentagonal bipyramidal geometry and possesses heptacoordination in THF, where five oxygen lone pairs coordinate equatorially and two iodine atoms binds axially to the samarium ion for stabilization [19]. The geometry of the samarium ion is altered with additives or ligands [10,11]. The samarium iodide shows octahedral geometry with the Lewis basic hexamethylphosphoramide (HMPA) in $[SmI_2(HMPA)_4]$, which elevates the reduction potential of $SmI_2$ through improved electron transfer [20]. Well-recognized proton sources such as water and alcohols can also increase the reduction potential of $SmI_2$ [21,22]. Thus, the oxyphilic nature and enhanced reduction potential of the samarium ion is accountable for stereoselective transformations [23].

SmI$_2$ has been generously utilized in reductive couplings for inter- and intramolecular carbon–carbon and carbon–heteroatom bond formation as well as functional group manipulation [24]. It is noteworthy to mention that SmI$_2$ mediated cross coupling reactions have led to widespread applications in the direct access to diverse complex molecular scaffolds [25]. The enhanced reduction potential of SmI$_2$ with HMPA allows the formation of various reactive intermediates leading to the access of complex architecture [26,27]. In this context, the enormous applications of SmI$_2$ in the key reaction step during the total synthesis of natural products is remarkable [28–32].

It is well-established that SmI$_2$ functions as mild reagent for various radical and ionic coupling reactions through reduction and reductive annulation involving aldol [33–35], Barbier [36,37], Grignard [38], reformats [39] and pinacol coupling [40,41] reactions as well as Diels–Alder dipolar addition [42]. In this regard, our group has reported various SmI$_2$ mediated reductive reactions of amines and imines [43–47].

Kagan and Molander performed pioneering research on applications of SmI$_2$ in various coupling reactions including asymmetric transformations [36,48–51]. It can be extensively used for generation of a chiral center, induced from an adjacent chiral auxiliary either connected to the substrate [52] or to the ligands [53]. They can be categorized either as enantioselective or diastereoselective asymmetric synthesis [54–56]. Moreover, being highly oxophilic, the chelating effect of Sm(II) or Sm(III) ions to more than one Lewis basic center in either the substrates, additives or solvents induces significant stereoselectivity in the products through the formation of distinct transition states and intermediates [57,58].

In this regard, a fair collection of reports in the literature on samarium-mediated organic transformations has been drafted in the form of reviews in the last ten years [10,11,59], whereas the asymmetric version [6] of these reactions has been rarely highlighted in the past few decades. The previous compilation on asymmetric reactions using SmI$_2$ by Kagan comprises the reports until 2008 [54]. Therefore, the urgency to update and critically analyze the overall asymmetric reactions has become mandatory. The present review summarizes the asymmetric reactions with catalytic and stoichiometric use of SmI$_2$ utilizing the adjacent chiral center in the substrate or a chiral ligand to induce asymmetry within the molecule.

## 2. Samarium-Mediated Ionic and Radical Asymmetric Reactions

### 2.1. Ketyl Radical-Olefin/Allene/Allyl Cyclization

2.1.1. Three-Membered Ring Forming Reactions

A unique example of a SmI$_2$-mediated 3-*exo*-trig cyclization of $\beta,\gamma$-unsaturated ketones and aldehydes was presented by Ortiz and Armesto in 2010 (Scheme 1) [60]. Treatment of the precursors with SmI$_2$-$^t$BuOH afforded functionalized cyclopropanols in good yields. A plausible mechanism involving the generation of a ketyl-type radical anion intermediate was hypothesized to explain the observed diastereoselectivity.

**Scheme 1.** Synthesis of cyclopropanols via 3-*exo*-trig cyclization and its mechanistic approach.

Furthermore, terminal alkenes substituted with activated aromatic groups, such as fluorene, indene and 4-cyanophenyl, were proven as appropriate substrates for the reaction (e.g., only alcohol and elimination products were observed using unsubstituted styrene), suggesting that the alternative mechanism involving reduction of the aromatic ring might be operative in this reaction.

### 2.1.2. Five-Membered Ring Forming Cyclization and Cascade Reactions

In 2017, Procter and co-workers developed an enantioselective SmI$_2$-mediated radical cyclization and cascade reaction via the formation of a ketyl radical with high enantio- and diastereo-control (Scheme 2) [61]. A recyclable chiral ligand and achiral alcohol as additives were utilized that can transform the symmetrical ketoester to a complex carbocyclic moiety with multiple stereocenters. The enantioselective desymmetrizing intramolecular cyclization of a ketyl radical with alkene and cyclization cascades in dienyl $\beta$-ketoesters was facilitated by an in situ-generated chiral Sm(II) reagent. Thereafter, the chelated Sm(III) ketyl intermediates generated from Sm(II) reagents accelerated the radical cyclization through desymmetrization of the dienyl $\beta$-ketoesters to form versatile mono- and polycyclic scaffolds containing a combination of multiple chiral centers with alkenyl units for further derivatization with excellent stereocontrol.

The dienyl $\beta$-ketoesters was purposefully introduced as dual-point-linking substrates to enhance the coordination with the Sm(II) reagent. The Lewis basic ester group in the substrate was utilized to coordinate to Sm(II) for promoting the reduction and directing the stereochemistry of samarium(III) ketyl cyclizations with the aid of chelated transition states. Furthermore, it was assumed that a multidentate chiral diol as the ligand coordinates

with Sm(II) to afford alteration in ionic radius for the generation of a Sm(III) intermediate, thereby following single electron transfer (SET). It induces the chirality in the Sm(III) ketyl radical cyclization and simultaneously serves as source of protons for the anions generated in the reaction.

**Scheme 2.** *Cont.*

**Scheme 2.** Samarium iodide-mediated radical cyclization and cascade reaction.

In 2011, Yang and Li proposed total synthesis of pseudolaric acid A from easily available starting compounds. The total synthesis was comprised of samarium-mediated intramolecular alkene-ketyl radical annulation and a ring-closing metathesis reaction as the key steps to stereoselectively form a pseudolaric acid A core, a rarely observed [5,7]-bicyclic scaffold with transfusion (Scheme 3). The unique synthetic strategy portrays a facile access to pseudolaric acids scaffolds, and possesses excellent potential to deliver a wide range of pseudolaric acid analogues. Although SmI$_2$-promoted ketyl radical cyclization had been widely utilized to construct structurally varied natural products and polycyclic scaffolds, the stereoselective version, however, remains less explored. The stereochemical fate of Sm-promoted cyclization exclusively depends on the nature of the substrate, additive and solvent. Based on this survey, the initial attempt to perform SmI$_2$-mediated alkene-ketyl radical annulation reactions in solvents, such as THF, acetonitrile and dimethyl ether (DME), led to low yields and poor diastereoselectivity. However, additives such as HMPA could enhance the reduction potential of SmI$_2$ and enhance both the yield and *trans* selectivity of the product.

**Scheme 3.** Samarium-mediated alkene-ketyl radical cyclization for the synthesis of pseudolaric acid A.

In 2010, Procter et al. revealed an unprecedented cascade cyclization following a sequence of reactions involving conjugate reduction, stereoselective aldol cyclization and chemoselective reduction of lactone mediated by $SmI_2$ and water (Scheme 4) [62]. The logically formulated strategy leads to the facile synthesis of densely substituted cyclopentanol derivatives with two vicinal stereoselective quaternary centers. The structurally complex and stereoselective cyclopentanol derivative was converted to a key intermediate for the asymmetric synthesis of stolonidiol. The reaction begins with reduction of a conjugated electrophilic alkene and further furnish the Sm-enolate intermediate. The stereoselective intramolecular aldol condensation with the suspended ketone furnished the spirocyclic cyclopentanol, where the lactone ring was reduced to triol, without affecting the acetate group. The stereochemistry of the predominant product is in synchronization with the hypothesized transition state, where both carbonyls from ester and keto functionalities coordinate and direct to Sm(III) in the proposed Sm(III)-enolate intermediate.

**Scheme 4.** Samarium-mediated cascade cyclization to cyclopentanol.

### 2.1.3. Seven-Membered Ring Forming Cyclization Reactions

In 2010, Reissig accentuated an important instance of $SmI_2$-mediated 7-*exo*-trig cyclization in *tert*-butanol ($^t$BuOH) to form benzannulated carbocycles (Scheme 5) [63]. An 8-*endo*-trig cyclization promoted by $SmI_2$-$^t$BuOH between a terminal unactivated alkene and ketone was explored initially. It was observed that the $\gamma$-ketoester with substituents at the $\beta$-styryl carbon cyclized to afford the 8-*endo*-trig cyclization, whereas cyclohexanone derived $\gamma$-ketoester dominated the 7-*exo*-trig instead of 8-*endo*-trig cyclization.

**Scheme 5.** Samarium iodide-mediated cyclization to benzannulated carbocycles.

## 2.2. Ketyl Radical-α,β Unsaturated Carbonyl Cyclization

In 2012, Hsu reported SmI$_2$-mediated reductive cyclization of enones to spiranes containing γ-hydroxyketones in good yields and with diastereoselectivity (Scheme 6) [64]. The reductive cyclization was found to give the best yields when two equivalents of SmI$_2$ were used with two equivalents of methanol as an additive in THF at 0 °C.

**Scheme 6.** Synthesis of Spirocycles via 5-*exo*-Trig and 6-*exo*-Trig Cyclizations of Enones.

Englerin A is a sesquiterpene, isolated from the bark of Phyllanthus engleri. It is an important inhibitor of the growth of human cancer cells. In 2011, Chain reported a synthetic strategy for (-)-Englerin A, where SmI$_2$ induced the ketone-enone 6-*exo*-trig cyclization reaction in the key step to afford the tricyclic precursor for the synthesis of the desired natural product (Scheme 7) [65]. It was realized that the SmI$_2$ cyclization proceeded smoothly in THF in the presence of HMPA at room temperature. The Michael

adducts generated in the previous step produced ketoalcohol **7a** in a reductive carbonyl-alkene cyclization to afford the product in 43% yield with excellent diasteroselectivity. The diastereoselectivity is attributed to the simultaneous coordination of oxygen atoms from the ketyl radical and the carbonyl with Sm (II) affording the Sm(III) radical intermediate **7A**. The cross-coupling of a ketyl radical with an olefin resulting in cyclization led to another radical intermediate, **7B**, which generated product **7a** in 43% yield and >95:5 diastereomeric ratio (*dr*).

**Scheme 7.** (**a**) Cyclization onto the enone in the total synthesis of (-)-Englerin A. (**b**) Proposed mechanism for cyclization.

In 2012, Adachi and Nishikawa reported a SmI$_2$-$^t$BuOH promoted 4-*exo*-trig ketone-enone cyclization to afford a highly strained cyclobutene ring as the interesting and challenging precursor (**8a**) for Solanoeclepin A (Scheme 8) [66]. This method can be considered as a straightforward approach to the direct construction of tricyclo [5.2.1.0$^{1,6}$]-decane scaffolds with good yields and excellent diastereoselectivity.

**Scheme 8.** 4-*exo*-trig cyclization for the synthesis of Solanoeclepin A.

In 2012, Zhai and coworkers executed a 15-step synthetic route to access Merrilactone A through a samarium-mediated cross-coupling reaction as one of the key steps (Scheme 9) [67]. In this synthetic method, 5-*exo*-trig cyclization of a ketone precursor proceeded with an excellent yield to afford the desired tetracycle precursor (**9a**) in 95:5 *dr* by treatment with SmI$_2$-THF at room temperature.

**Scheme 9.** 5-*exo*-trig cyclization in the total synthesis of Merrilactone A.

(-)-GB 13, a class of alkaloid which was isolated from the bark of Galbulimima belgraveana, has gained enormous attention from the pharmaceutical industry. In 2010, Ma reported the total synthesis of alkaloid (-)-GB 13 by a samarium iodide-mediated 5-*exo*-trig reductive cross coupling reaction of a ketone and an enone with a good yield of the single diastereomer [68]. The occurrence of a coupling reaction involving a radical anion intermediate was hypothesized which might have been initiated via a single electron transfer to the ketone by SmI$_2$ (Scheme 10).

**Scheme 10.** 5-*exo*-Trig Cyclization in the Total Synthesis of (-)-GB 13 and the plausible mechanism.

During the formation of the [3.2.1]-bicyclic core structure, a 1,3-diaxial connection was favored among all other possibilities. Consequently, the adversative diaxial conformation **10B** was preferred over **10A** for SmI$_2$-mediated carbonyl–alkene reductive coupling for the generation of a [3.2.1]-bicyclic system. The reaction required increased temperatures to deliver the thermodynamically less stable diaxial conformer.

The groups of Sabitha and Takahashi independently reported a cross-coupling strategy for the synthesis of the tetrahydropyran counterpart of aspergillide A, which possesses significant cytotoxicity [69]. One of the key steps of this synthesis included treatment of the respective aldehydes with SmI$_2$-MeOH, generating the tetrahydropyran ring system (**11a**) with excellent yield and diastereoselectivity. The formal total synthesis of aspergillide A was completed by Takahashi in four steps (Scheme 11).

**Scheme 11.** SmI$_2$-MeOH mediated formation of the tetrahydropyran ring in aspergillide A.

Honda et al. introduced an effective stereoselective synthesis of (-)-stemoamide in 2011 [70]. The intramolecular cross-coupling reaction of an in situ-generated ketyl radical intermediate from an aldehyde with the $\alpha,\beta$-unsaturated ester by $SmI_2$-promoted 7-*exo*-trig cyclization to obtain the tricyclic precursor for the alkaloid (-)-stemoamide after lactonization (**12a**) in 60% yield with good diastereoselectivity (Scheme 12).

In the absence of concrete evidence clarifying the reaction mechanism, the stereoselectivity obtained could be explained with the conjecture that HMPA promotes the reaction to attain a sterically favored diradical transition state (TSA), where two substituents were situated in a diequatorial-like orientation which is energetically favorable. The presence of HMPA in the reaction mixture induces a strong redox potential in samarium diiodide to generate diradicals as the key intermediate.

**Scheme 12.** Synthesis of (-)-stemoamide via 7-*exo*-trig cyclization.

## 2.3. Ketyl Radical-$\alpha,\beta$ Unsaturated Sulfonyl Cyclization

In 2010, Nakata et al. reported the reductive 6-*exo*-trig cyclizations of (*Z*)- and (*E*)-$\beta$-alkoxyvinyl sulfones with aldehydes using polycyclic ethers to afford 2,6-*syn*-2,3-*cis*-tetrahydropyran-3-ol and 2,6-*syn*-2,3-*trans*-tetrahydropyran-3-ol (Scheme 13) [69]. These polyethers were the key intermediate in synthesis of gymnocin-A by using the Suzuki–Miyaura reaction. The reaction of the (*Z*)-isomers was found to be slightly less efficient than that of the corresponding (*E*)-olefins; it provided *cis*-fused polyethers with excellent diastereocontrol. The authors proposed that the observed decrease in efficiency resulted from reduced chelation to the Sm(III) ketyl in the intermediate state (Scheme 13b).

It was hypothesized that in the (*E*)-isomer, the $SmI_2$-mediated reduction of the aldehyde via single-electron transfer delivers a ketyl radical, **13A**, followed by intramolecular C–C bond formation involving chelation with Sm(III) and sulfone to give **13B**. The single electron transfer from the second equiv. of $SmI_2$ reduces the radical to an anion, which after abstraction of a proton from MeOH, gave 2,6-*syn*-2,3-*trans*-**13b**. On the contrary, 2,6-*syn*-2,3-*cis*-**13b** could be the predominant product from the (*Z*)-isomer **13a′** through the chelated transition states **13A′** and **13B′**.

**Scheme 13.** (**a**) Reductive cyclization of (*Z*)-vinylsulfones and (*E*)-vinylsulfones with aldehydes. (**b**) Mechanistic pathway for cyclization.

## 2.4. Amino Ketyl Radical Olefin Cyclization

In 2015, Szostak developed a new stereoselective method for the generation of five- and six-membered imide moieties from an unactivated alkene via the chemoselective formation of aminoketyl radicals, which is an important intermediate [71]. The addition of the aminoketyl radical formed through electron transfer from the five- and six-membered imide to the alkene was identified as the rate determining step (RDS) in the reaction (Scheme 14). Moreover, the stability of the aminoketyl radicals due to the $n_N \rightarrow$ SOMO (singly occupied molecular orbital) conjugation accounts for the excellent diastereoselectivity (>95:5). Notably, neither the overreduction of the amide group nor the aminoketyl was observed. This reductive desymmetrization of five- and six-membered imides is performed by $SmI_2$/water via a two electron transfer where the water complexes with Sm(II) and acts as the proton source. The proposed ketyl radical anion (**14A**) was assumed to be possible in the presence of water as a ligand, as it is capable of binding in the inner coordination sphere of Sm(II) and acting as proton source, as evident from previous reports. This method opens a new avenue in carbon-carbon coupling reactions along with an upgrade in the concept of the ketyl radical through the generation of an umpolung aminoketyl synthon.

**Scheme 14.** Samarium-mediated reductive desymmetrization of cyclic imides.

Sequential and Dearomatizing Cyclization Reactions

$SmI_2$ has been well-recognized as functioning as an electron transfer reductant. The synthesis of quaternary stereocenters, present in biologically important natural products and drugs, possesses a significant challenge due to their structural complexity. Therefore, a cascade cyclization reaction involving a selective radical reaction with $SmI_2$ has made the generation of quaternary stereocenters feasible. In 2017, Procter et al. described the reduction of amide type carbonyls through single electron transfer using $SmI_2$-mediated cascade cyclisation to construct pharmaceutically important tricyclic barbiturates with high diastereocontrol [72]. The mechanistic interpretation of the sequential cascade radical cyclization leading to significant diastereoselectivity was proposed and is shown in Scheme 15. It was hypothesized that the single electron transfer from the Sm(II) catalyst to the carbonyl of amide resulted in a Sm(III)-coordinated radical intermediate (**15A**). Further ring closure via 6-*endo*-trig cyclization through the formation of a stable chair conformation (**15B**) produced the quarternary stereocenter of the hemiaminal intermediate (**15C**), which was quenched with acid to generate the polycyclic enamine (**15D**) which further generated the product **15a** through rearrangement with loss of proton. The pseudoaxial orientation of the larger substituent in the alternative chair and boat conformations disfavor product formation.

**Scheme 15.** SmI$_2$-mediated cascade cyclisation of an alkene with an amino ketyl radical.

Samarium was successfully employed for the construction of complex polycyclic scaffolds by Procter and coworkers in 2016 [72]. The protocol utilizes dearomatizing radical cyclization of the radical generated by the reduction of an amide type carbonyl via single electron transfer followed by cascade annulation. SmI$_2$–H$_2$O–LiBr portrays a unique and efficient reagent system for this purpose of accessing spiropolycyclic scaffolds with multiple stereocenters possessing significant diastereoselectivity (Scheme 16). In the dearomatizing single electron transfer radical formation and cascade cyclization, Sm(II) undergoes transfer of an electron to afford a ketyl radical of the amide-type carbonyl (**16A**). The 5-*exo*-trig cyclization of the ketyl radical further generates another radical intermediate (**16B**). Subsequent spirocyclization of the radical intermediate occurs through

a chair conformation, where the dipole–dipole interactions were reduced, and produces the final product after single electron transfer reduction and protonation.

**Scheme 16.** SmI$_2$-mediated dearomatizing radical cyclization forming spiropolycycles.

### 2.5. Ester Ketyl Radical-Allene/Olefin Cyclization and Cascades

In 2012, Procter exemplified a SmI$_2$–H$_2$O mediated reductive radical cyclization of unsaturated lactones to afford substituted cycloheptanes with adjoining stereocenters in high yields and diastereoselctivities [73]. The stereocontrol in product formation post-cyclization was attained with a relay around the ring which serves as the key to the observed high diastereoselectivities. When a combination of two alkenes, an alkene and an alkyne, or an allene and an alkene, exists in the substrate, the probability of dual radical anion intermediates leads to radical cascade cyclizations. This enables easy access to complex architectures in a single pot, using an easily accessible reagent with a maximum of four adjacent stereocenters.

The cyclizations proceeded by the trapping of radical anions formed by the electron transfer reduction of the lactone carbonyl. It was speculated that the single electron transfer from SmI$_2$–H$_2$O to the ester carbonyl favors an axial radical anion (**17A**) which terminates through cyclization. The ring cleavage in the hemiacetal (**17B**) by reduction and proton abstraction results in an enone (**17C**). Thereafter, a second radical anion (**17D**) is generated through reduction of the enone, which gets selectively protonated at the $\beta$-position. Samarium(III) enolate (**17E**) was produced through consecutive reduction, and that further proceeds with diastereoselective $\alpha$-protonation to produce (**17F**). Ultimately, the cycloheptanone intermediate (**17F**) was reduced to access the product with high diastereocontrol via a third radical anion (**17G**) (Scheme 17). Thus, the sequence of intermediates facilitates the passing of chirality from carbon to carbon in the cyclized product.

**Scheme 17.** Reductive radical cyclization of unsaturated lactone to substituted cycloheptanes.

Using Meldrum's acid template, in 2012 Procter published a SmI$_2$-mediated 5-*exo*-trig/5-*exo*-trig and 5-*exo*-trig/6-*exo*-trig cascade cyclization leading to complex bicyclic alcohols containing four adjacent stereocenters [74]. The differential activation of the olefin acceptors allows the high chemoselectivity in the cascade sequences (Scheme 18). In one extreme case, the olefin acceptors bearing 4-Br-C$_6$H$_4$ and -Ph substituents participated in a

fully chemo- and diastereoselective cyclization, which was initiated by a selective cross-coupling of the ketyl radical with the 4-Br-C$_6$H$_4$-substituted olefin (-Ph). The cyclization selectivity is consistent with the relative stabilization of the produced carbon-centered radicals by these substituents. The mechanism of the reaction was predicted to proceed with cyclization of the first radical anion (**18A**) to afford a ketone intermediate (**18B**) with high diastereoselectivity through a preferred *anti*-transition state. Subsequent electron transfer from SmI$_2$–H$_2$O generated another radical anion (**18C**), quenching the remaining acceptor through the energetically preferred *anti*-transition state to deliver annulated products with high diastereoselectivity.

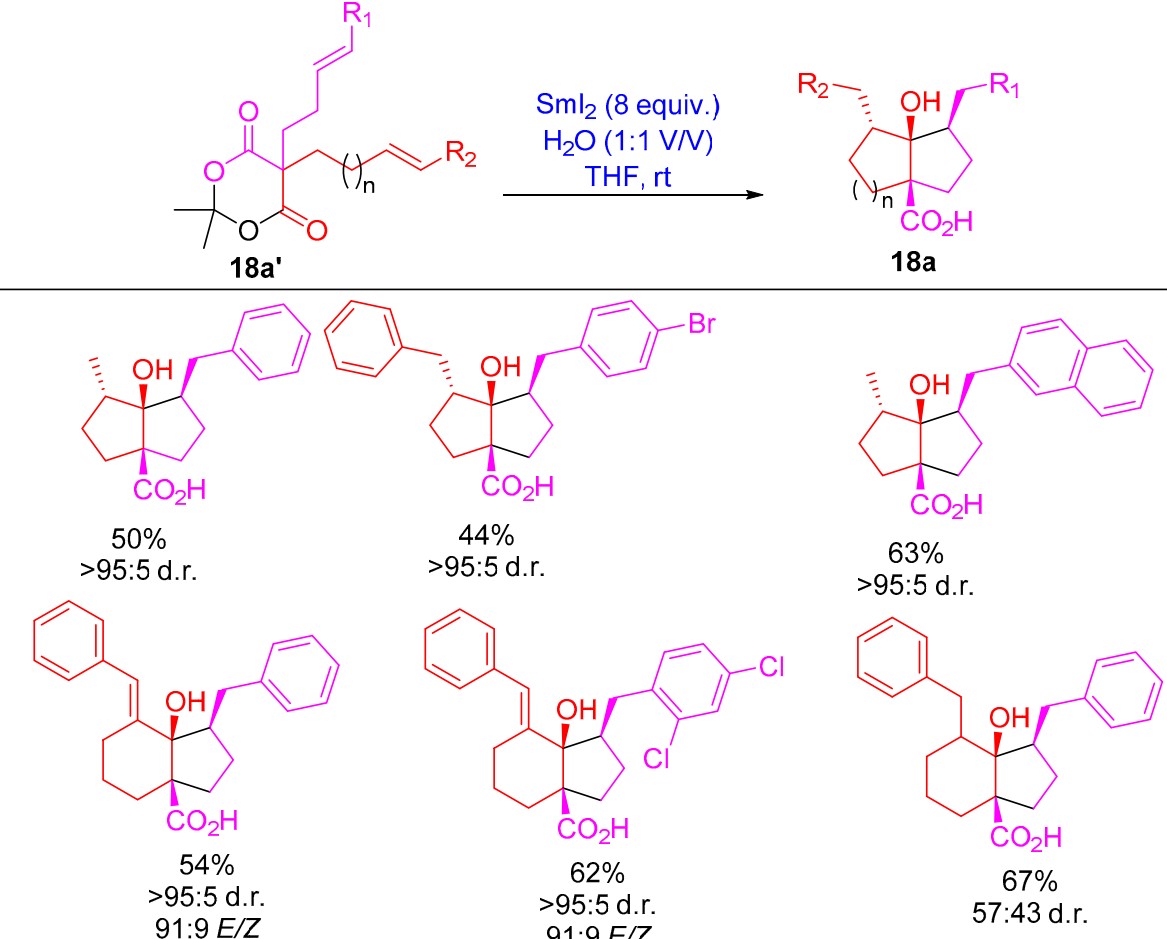

**Scheme 18.** *Cont.*

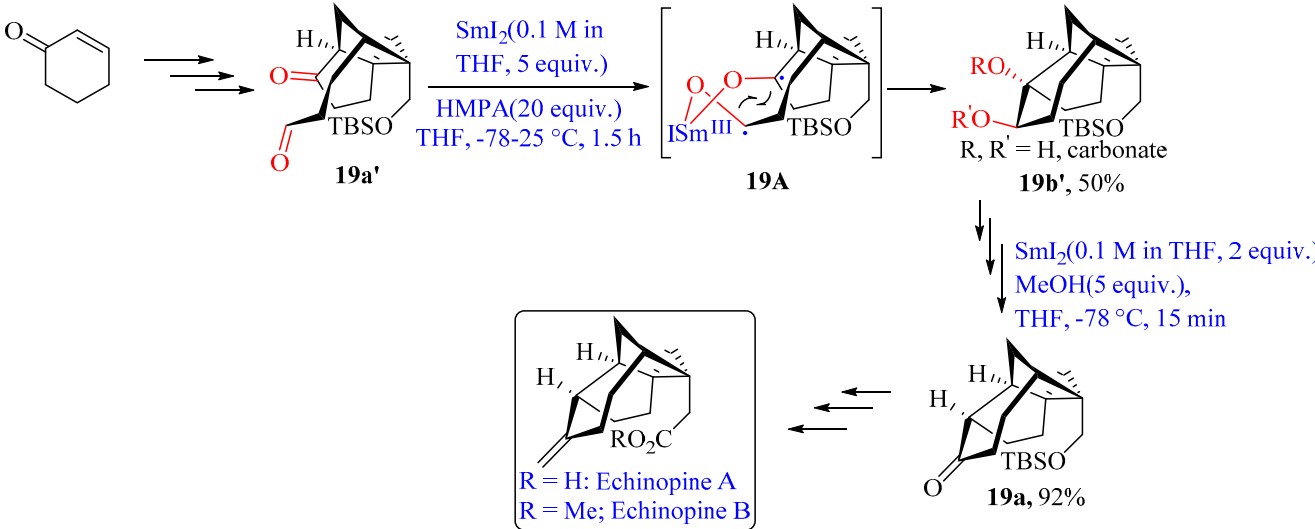

**Scheme 18.** SmI$_2$-mediated 5-*exo*-trig/5-*exo*-trig and 5-*exo*-trig/6-*exo*-trig cascade cyclization.

### 2.6. Pinacol-Type Radical Cyclizations

### 2.6.1. Ketyl-Carbonyl Diradical Cyclizations

Samarium diiodide has been extensively employed to facilitate the total synthesis of various natural products. In this context, Echinopines A and B, possessing a unique [3.5.5.7] framework, have been produced in both racemic as well as enantiomerically pure forms. The total synthesis of Echinopines A and B involved a novel intramolecular rhodium-catalyzed cyclopropanation and a -SmI$_2$ mediated ring closure through pinacol coupling, which were the key steps to produce a single diastereomer, as reported by Nicolaou and Chen in 2010 [75]. The stereoselectivity of the product from the SmI$_2$-promoted reaction was supported by hypothesizing a cyclic chelated transition state with Sm(III) (Scheme 19).

**Scheme 19.** SmI$_2$-mediated ring closure through pinacol coupling in the synthesis of Echinopine A and B.

### 2.6.2. Ketyl Radical-Carbonyl Cyclizations

Wu and coworkers reported an asymmetric total synthesis of (+)-caldaphnidine J, which is a yuzurimine-type alkaloid, in 2020 using a Sm-mediated pinacol reaction strategy

to produce one of the key diol products (**20a**) [76]. The synthetic pathway consisted of several steps including a Pd-catalyzed regioselective hydroformylation that resulted in the critical aldehyde motif, and the construction of the 7/5 bicyclic system using a SmI$_2$-promoted intramolecular asymmetric pinacol coupling reaction and a one-pot Swern oxidation/ketene dithioacetal Prins reaction (Scheme 20).

**Scheme 20.** Sm-mediated pinacol reaction strategy in the synthesis of (+)-caldaphnidine J.

Burtoloso reported the SmI$_2$-mediated coupling of $\alpha$-amino aldehydes and ketones with methyl acrylate to form $\gamma$-aminomethyl-$\gamma$-butyrolactones (**21a**) in high yields and diastereoselectivities (Scheme 21). Extensive optimization revealed that H$_2$O serves as the optimal additive for this reaction, while use of MeOH and $^t$BuOH afforded lower yields. Interestingly, reductive cleavage of the C-N bond was not observed under these conditions. The methodology provides a concise access to indolizidine and quinolizidine alkaloids from $\alpha$-amino acids.

**Scheme 21.** SmI$_2$-mediated coupling of $\alpha$-amino aldehydes and ketones with methyl acrylate.

## 2.7. Barbier-Type $\pi$-Allyl Radical Cyclizations

In a recent report by Morimioto et al. in 2022 [77] on asymmetric total synthesis of sesquiterpenoid, toxicodenane was introduced, which has been identified as acting as an antidiabetic. The key step in the total synthesis involved a SmI$_2$-mediated Barbier-type annulation reaction generating the stereoselective bicyclic product, **22a**, in 99% yield with a *cis* configuration (Scheme 22). The formation of the $\pi$-allyl samarium intermediate with a chair conformation (**22B**) was responsible for dictating the diastereoselectivity

of the product (**22a**). Furthermore, stereoselective product was obtained through site selective allylic oxidation and dehydrative cyclization. The final compounds (+)- and (−)-toxicodenane displayed significant cell-protective effects against lipotoxicity-mediated inflammation and fibrosis.

**Scheme 22.** SmI$_2$-mediated Barbier-type annulation.

### 2.8. Sm(III) Relay Catalytic Three-Component Tandem [4 + 3]-Cycloadditions

A tandem three component reaction was introduced by Feng and coworkers via a [4 + 3] cycloaddition reaction of carbonyl ylide generated from α-diazoacetates (**23c′**) and aldehydes (**23b′**) with β,γ-unsaturated ketoesters to produce asymmetric 4,5-dihydro-1,3-dioxepines in up to 97% yield, with 99% *ee* (Scheme 23) [78]. The asymmetric synthesis was catalyzed by the bimetallic Rh(II) and Sm(III) salts in presence of *N,N′*-dioxide as the ligand. A plausible mechanism of the stereoselective induction was predicted through control experiments and X-ray crystal structures. The stereoselectivity induced from the *N,N′*- dioxide–Sm(III) Lewis acid catalyst was demonstrated by the crystal structure of the Sm(III) complexes of chiral *N,N′*-dioxides L$_4$-PrPr$_2$. A reaction sequence was proposed based on control experiments and absolute configuration of the products. It was assumed to proceed through the initial generation of a carbonyl ylide from α-diazoacetate and an aldehyde with the assistance of achiral Rh(II). The possible coordination of the carbonyl ylide intermediate **23A** with chiral Sm (III)–*N,N*-dioxide was supported by performing an operando IR experiment. The activation of the β,γ-unsaturated α-ketoester with Sm(III)/L$_4$-PrPr$_2$ through bidentate binding (**23C**) is possible due to the properties of rare earth elements to accommodate high coordination number.

The steric crowd of the 2,6-$^i$Pr$_2$C$_6$H$_3$ counterpart in the chiral *N,N′*-dioxide is responsible for the inaccessibility of the β-*Re* face of the unsaturated ketoester, promoting facile β-*Si* face attack by the keto ylide (**23A**) in endo fashion with its *Re–Re* face to yield the major enantiomer.

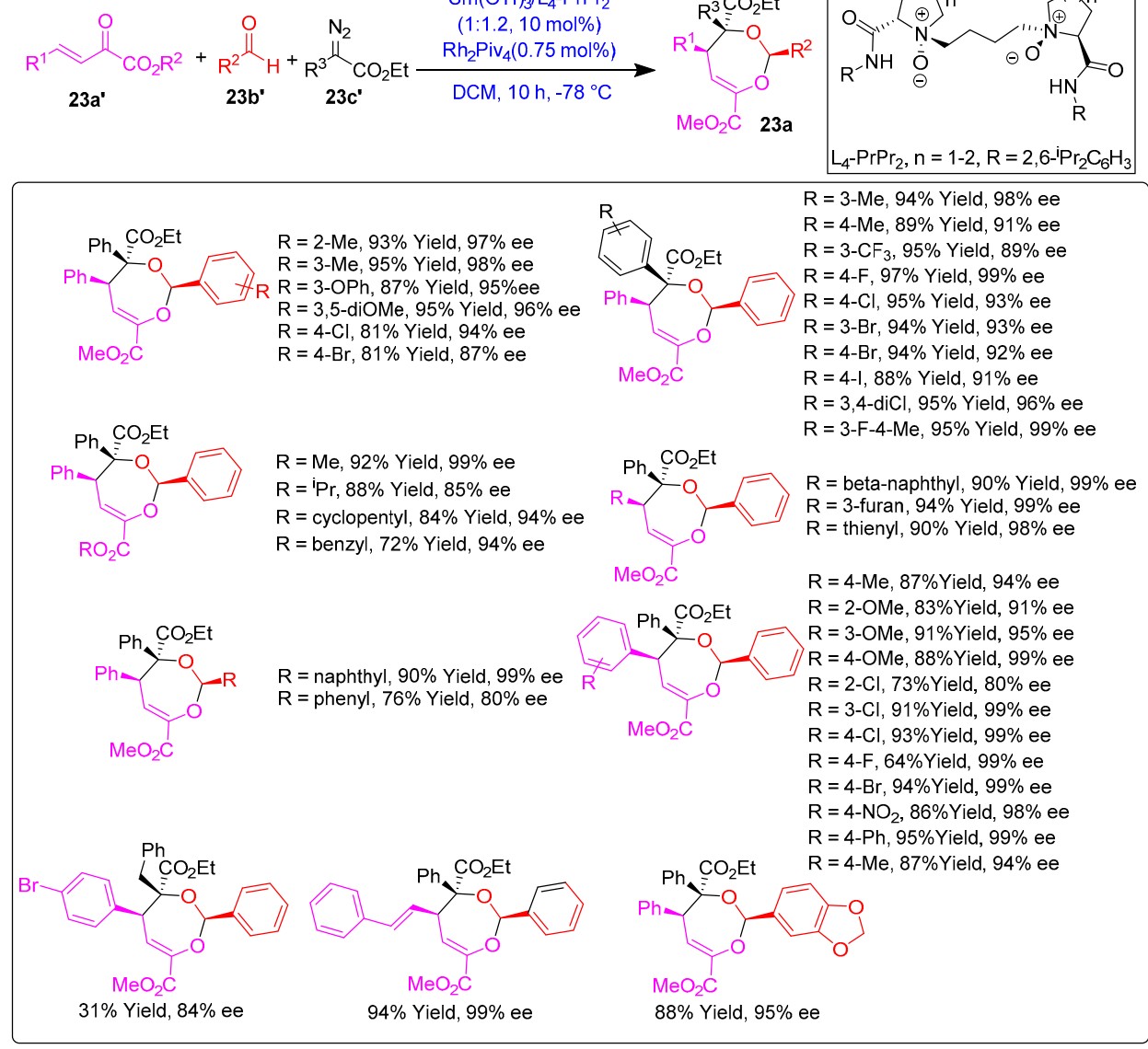

**Scheme 23.** *Cont*.

**Scheme 23.** [4 + 3] cycloaddition reaction of carbonyl ylide generated from α-diazoacetates and aldehydes.

## 2.9. Stereoselective Reduction of Ketone

In 2021, Liu and Fu et al. established the asymmetric total synthesis of rumphellclovane E, a clovane type sesquiterpenoid, by the several key steps including Rh-catalyzed cyclopropanation, Fe-catalyzed intramolecular reductive aldol reaction and samarium iodide-mediated stereoselective reduction of ketone (Scheme 24) [79]. The stereoselective reduction of ketone was done by a solution of $SmI_2$ in THF with good yield.

The reaction was assumed to proceed through the formation of ketyl radical intermediate (**24A**). It was proposed that the $SmI_2$ transfer an electron to the C2 carbonyl which then accommodates a proton from water to form the ketyl radical (**24A**) in which the carbon radical was stabilized as axial position through minimization of steric interaction with the axial methyl substitution at C4. The Sm(II) which is coordinated to C12 ketone might approach ketyl radical from axial position to form organosamarium(III) (**23B**), inspite of the steric hindrance of the axial methyl group at C4. The organosamarium(III) intermediate (**23B**) prefers Re face to accept proton and generate the desired product **24a**.

**Scheme 24.** SmI$_2$ mediated stereoselective reduction of ketone.

### 2.10. Chiral Ligand Promoted Hydroamination of Cyclopropene

Sm-catalyzed reactions could also be extended for stereoselective hydroaminations to generate chiral aminocyclopropanes. The existence of the aminocyclopropanes in pharmaceutically active natural products has led to the investigation of efficient synthetic route in the formation of this class of compounds. In 2016, Hou and co-workers established an atom economic synthetic strategy for the successful generation of chiral aminocyclopropanes (**25a**) via Sm-catalyzed intermolecular hydroamination reaction of cyclopropene derivatives with various substituted amines (Scheme 25) [80]. A chiral half sandwiched Sm-catalyst successfully delivered the enantioenriched addition products in high yields. The catalytic activity was found to be directly proportional to the impact of metal ion and the binding ligand where the perfect combination of these two can lead to fine selectivity and enhanced efficiency of catalyst and modify the reaction environment.

**Scheme 25.** *Cont.*

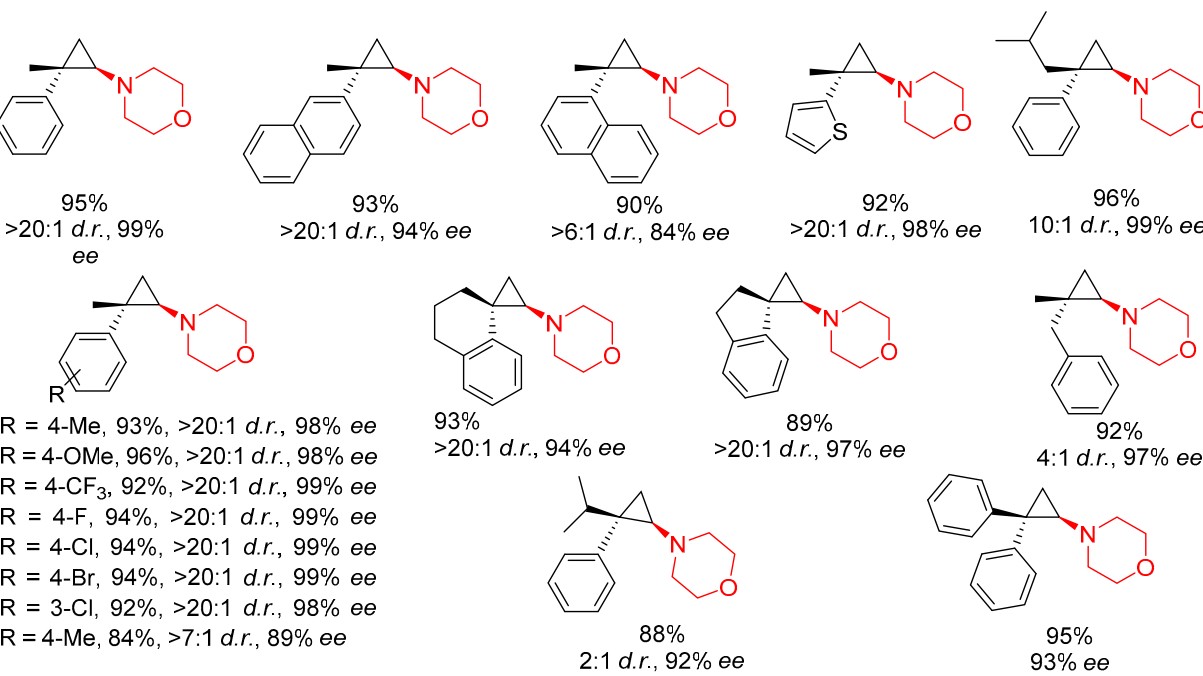

**Scheme 25.** Synthesis of chiral aminocyclopropanes by hydroamination of cyclopropenes.

### 3. Conclusions

In conclusion samarium-mediated/catalyzed asymmetric synthesis has widespread applications in organic transformations. It embraces facile access to complex architecture occurring in synthetic and natural products. The user-friendly nature of samarium reagents has led to their wide applicability. It possesses moderate to high chemo-, regio, enantio- and diastereoselectivity in various reactions through ionic and radical mechanisms. Samarium reagents have been employed as one and two electron transfer agents for the generation of radical or anions in various asymmetric synthesis. The erratic role of additive, ligand, and solvents in providing extra stability to the samarium for generation of reactive intermediates through effective coordination, efficiently dictates the selectivity of the products. Further, substantial exploration of catalytic activity of new samarium reagents needs to be done to establish effective reaction protocols since the catalytic reactions of samarium has been less documented till date. Samarium reagent system has been extensively employed in several cyclization and complex cascades along with reduction and intermolecular coupling reactions while the asymmetric transformations are yet to be epitomized the development of robust and efficient catalytic systems with samarium for intra and intermolecular asymmetric coupling reactions are the major challenges to be addressed.

Undoubtedly, the versatility and broad-spectrum applicability of Samarium reagents will continue to attract and indulge the scientific community to unveil several mysteries of impending synthetic transformations in coming years.

**Funding:** The article received no external funding.

**Institutional Review Board Statement:** Not applicable.

**Informed Consent Statement:** Not applicable.

**Data Availability Statement:** Not applicable.

**Acknowledgments:** Bimal Krishna Banik is grateful to US NIH, US NCI and the Kleberg Foundation of USA for the support of this research. Devalina Ray & Suman Majee is thankful to DST-SERB (CRG/2019/002333) for financial support.

**Conflicts of Interest:** The authors declare that they have no known competing financial interest or personal relationships that could have appeared to influence the work reported in this paper.

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
