# Peer review of "Samarium-Mediated Asymmetric Synthesis"

_catalysts, doi:10.3390/catal13010024_

Round 1

Reviewer 1 Report

The review by Majee, Ray and Banik covers samarium-mediated transformations from the last 12 years. The authors provide a balanced overview of the field and the review should be suitable for the readership of the present journal. Nevertheless, further work on the English is required before publication for a good flow. In addition, the Conclusions section is currently more of a summary and would need to be expanded and highlight emerging trends and future opportunities in the field.

Author Response

Response to Reviewer 1 Comments:

Point 1: The review by Majee, Ray and Banik covers samarium-mediated transformations from the last 12 years. The authors provide a balanced overview of the field and the review should be suitable for the readership of the present journal. Nevertheless, further work on the English is required before publication for a good flow. In addition, the Conclusions section is currently more of a summary and would need to be expanded and highlight emerging trends and future opportunities in the field.

Response 1: Thanks for the comments. The modifications has been incorporated as per suggestion. Further conclusion section has been expanded with the present trend, challenges and future scope of Sm-mediated/.catalyzed asymmetric reactions.

Reviewer 2 Report

The samarium mediated/catalyzed asymmetric synthesis has widespread applications in organic transformations. There are some reviews focusing on the samarium mediated organic transformations in the past decade, however, the asymmetric version of these reactions has been rarely highlighted. In this review, the authors have comprehended the developments in samarium mediated/catalyzed asymmetric organic synthesis of past 12 years. Although many transformations described in the manuscript are not catalytic, this review should be interesting to the synthetic community. The review is well organized and written, and therefore recommended for the publication in Catalysts after a minor revision as follows,

1)       Scheme 2 should be cited in the text.

2)       236th line, “Nakata et all” should be “Nakata et al”;

3)       In Scheme 13 (a), the structure of 13a is the same as that of 13b; The number of 13a and 13 b in Scheme 13b is incorrect as their structures are different with those in Scheme 13a;

4)       In Scheme 14 (271st line), the structure of 14 a is incorrect;

5)       387th line, “stereoselective bicyclic product 24 in 99% yield” should be “stereoselective bicyclic product 22a in 99% yield”;

6)       The number of 23a’, 23 a, 23A, 23B should be 24a’, 24 a, 24A, 24B….

Author Response

Response to Reviewer 2 Comments:

The samarium mediated/catalyzed asymmetric synthesis has widespread applications in organic transformations. There are some reviews focusing on the samarium mediated organic transformations in the past decade, however, the asymmetric version of these reactions has been rarely highlighted. In this review, the authors have comprehended the developments in samarium mediated/catalyzed asymmetric organic synthesis of past 12 years. Although many transformations described in the manuscript are not catalytic, this review should be interesting to the synthetic community. The review is well organized and written, and therefore recommended for the publication in Catalysts after a minor revision as follows,

Point 1: Scheme 2 should be cited in the text.

Response 1: As per reviewer’s comment, Scheme 2 has been cited in the text.

Point 2: 236thline, “Nakata et all” should be “Nakata et al”;

Response 2: Thanks for the observation, we have corrected it in the text.

Point 3: In Scheme 13 (a), the structure of 13a is the same as that of 13b; The number of 13a and 13 b in Scheme 13b is incorrect as their structures are different with those in Scheme 13a;

Response 3: As per reviewer’s suggestion, the corrections have been incorporated in the schemes.

Point 4: In Scheme 14 (271stline), the structure of 14 a is incorrect;

Response 4: The structure of 14a in scheme 14 has been corrected.

Point 5: 387thline, “stereoselective bicyclic product 24 in 99% yield” should be “stereoselective bicyclic product 22a in 99% yield”;

Response 5: The correction has been implemented according to reviewer’s suggestion.

Point 6: The number of 23a’, 23 a, 23A, 23B should be 24a’, 24 a, 24A, 24B….

Response 6: Thanks for the comments. Changes has been made in the manuscript accordingly.

Reviewer 3 Report

This review entitled Samarium Mediated Asymmetric Synthesis, mainly covers Samarium mediated ionic and radical asymmetric reactions. This review is then organized in various subsections such as i) Ketyl radical-olefin/allene/allyl cyclisations; ii) ketyl radical a,b-unsaturated carbonyl cyclization; iii) ketyl radical a,b-unsaturated suylfonylyl cyclization; iv) Amino Ketyl radical-olefin cyclization; v) estern ketyl radical-alene/olefin cyclization and cascade; vi) pinacol-type radical cyclization; vii) Sm(III) relay catalytic three-component tándem [4+3]-cycloaddition; viii) Stereoselective reduction of ketone; and ix)Chiral ligand promoted hydroamination of cyclopropene.

In general terms, this review covers some examples described in literature where the use of samarium salts allowed to access different compounds in a enantio- or stereoselective fashion. Regarding this manuscript I would say that needs a deep polish of the english since many typographical mistakes have been found. I also miss some papers that could give some value to the review. For example I think that next papers about asymmetrical synthesis employing samarium should be also included prior to publication (Eur. J. Org. Chem. 2013, 4953–4961; Chem. Commun. 2004, 2502–2503; Organic. Biomol. Chem. 2005, 3, 1435-1447)

Regarding typographical mistakes here are a few examples…

Lines:

31.-  in situ

36.- It is mentioned samarium diiodide such a catalyst. Sometimes it works on a stoichiometric or substoichiometric manner.

235 and 254.- have the same numeration for both sections

206.-  1,3-diaxial

209.- bicyclic system

225.- (-)-stemoamide

238.- and throuh all the paper… syn/anti/cis/trans/exo/endo/Si/Re/E/Z…) should be written in italics.

295.- Procter

In summary, this piece of paper deserves publication but needs to check and polish english before publication. Also some interesting related literatura should be included.

Author Response

Response to Reviewer 3 Comments:

Point 1: 

This review entitled Samarium Mediated Asymmetric Synthesis, mainly covers Samarium mediated ionic and radical asymmetric reactions. This review is then organized in various subsections such as i) Ketyl radical-olefin/allene/allyl cyclisations; ii) ketyl radical a,b-unsaturated carbonyl cyclization; iii) ketyl radical a,b-unsaturated suylfonylyl cyclization; iv) Amino Ketyl radical-olefin cyclization; v) estern ketyl radical-alene/olefin cyclization and cascade; vi) pinacol-type radical cyclization; vii) Sm(III) relay catalytic three-component tándem [4+3]-cycloaddition; viii) Stereoselective reduction of ketone; and ix)Chiral ligand promoted hydroamination of cyclopropene.

In general terms, this review covers some examples described in literature where the use of samarium salts allowed to access different compounds in a enantio- or stereoselective fashion. Regarding this manuscript I would say that needs a deep polish of the english since many typographical mistakes have been found. I also miss some papers that could give some value to the review. For example I think that next papers about asymmetrical synthesis employing samarium should be also included prior to publication (Eur. J. Org. Chem. 2013, 4953–4961; Chem. Commun. 2004, 2502–2503; Organic. Biomol. Chem. 20053, 1435-1447)

Response 1: As per reviewer's suggestion the grammar, language, typographical errors have been cross checked and corrected accordingly. The references have been updated with the papers suggested by the reviewer as 55 and 56 ref. However, the ref. Eur. J. Org. Chem. 2013, 4953–4961 seems to be different focusing on the reaction with chromium.

Point 2: Regarding typographical mistakes here are a few examples…

Lines:

31.-  in situ

36.- It is mentioned samarium diiodide such a catalyst. Sometimes it works on a stoichiometric or substoichiometric manner.

235 and 254.- have the same numeration for both sections

206.-  1,3-diaxial

209.- bicyclic system

225.- (-)-stemoamide

238.- and throuh all the paper… syn/anti/cis/trans/exo/endo/Si/Re/E/Z…) should be written in italics.

295.- Procter

Response 2: 31.-  changed to in situ

36.- samarium diiodide has been mentioned to act in catalytic as well as stoichiometric amount.

235 and 254.- the numeration has been changed in both sections

206.-  Changed to 1,3-diaxial

209.- hanged to bicyclic system

225.- Changed to (-)-stemoamide

238.- syn/anti/cis/trans/exo/endo/Si/Re/E/Z has been written in italics.

295.- Changed to Procter

Point 3: In summary, this piece of paper deserves publication but needs to check and polish english before publication. Also some interesting related literature should be included.

Response 3: The language has been modified as required and related literature is included as per suggestion.

Reviewer 4 Report

To authors:

11.    Please use more color in the schemes.

22. Please use same color for arrows in the schemes (somewhere red, somewhere purple, …). And follow this rule for other issues.

33.   Some schemes have been brought in rectangular space, some not. Please organize all in the same way.

44.       Please bring the list of all abbreviation used in the manuscript.

55.       Quality of some of the schemes are low, please check and redraw.

66.       Size of the rings and atoms in all schemes must be the same. So, it must be corrected.

Author Response

Response to Reviewer 4 Comments:

Point 1: Please use more color in the schemes.

Response 1: As per reviewers suggestion, all schemes have been represented with colors.

Point 2: Please use same color for arrows in the schemes (somewhere red, somewhere purple, …). And follow this rule for other issues.

Response 2: The uniformity in color has been maintained throughout the manuscript as advised.

Point 3: Some schemes have been brought in rectangular space, some not. Please organize all in the same way.

Response 3: All the schemes have been reorganized.

Point 4: Please bring the list of all abbreviation used in the manuscript.

Response 4: The abbreviations such as hexamethylphosphramide (HMPA), diastereomeric ratio (dr), Tetrahydrofuran (THF), Trimethylsilyl trifluoromethanesulfonate (TMSOTf), Rate determining step (RDS), Single occupied molecular orbital (SOMO) etc has been listed in manuscript.

Point 5: Quality of some of the schemes are low, please check and redraw.

Response 5: The schemes not drawn properly has been redrawn.

Point 6: Size of the rings and atoms in all schemes must be the same. So, it must be corrected.

Response 6: The corrections related to uniformity of rings and atoms in schemes has been incorporated.

Round 2

Reviewer 3 Report

Publish as is